# Neural Lyapunov Control

**Ya-Chien Chang**
UCSD
yac021@eng.ucsd.edu

**Nima Roohi**
UCSD
nroohi@eng.ucsd.edu

**Sicun Gao**
UCSD
sicung@eng.ucsd.edu

## Abstract

We propose new methods for learning control policies and neural network Lyapunov functions for nonlinear control problems, with provable guarantee of stability. The framework consists of a learner that attempts to find the control and Lyapunov functions, and a falsifier that finds counterexamples to quickly guide the learner towards solutions. The procedure terminates when no counterexample is found by the falsifier, in which case the controlled nonlinear system is provably stable. The approach significantly simplifies the process of Lyapunov control design, provides end-to-end correctness guarantee, and can obtain much larger regions of attraction than existing methods such as LQR and SOS/SDP. We show experiments on how the new methods obtain high-quality solutions for challenging robot control problems such as path tracking for wheeled vehicles and humanoid robot balancing.

## 1 Introduction

Learning-based methods hold the promise of solving hard nonlinear control problems in robotics. Most existing work focuses on learning control functions represented as neural networks through repeated interactions of an unknown environment in the framework of deep reinforcement learning, with notable success. However, there are still well-known issues that impede the immediate use of these methods in practical control applications, including sample complexity, interpretability, and safety [5]. Our work investigates a different direction: Can learning methods be valuable even in the most classical setting of nonlinear control design? We focus on the challenging problem of designing feedback controllers for stabilizing nonlinear dynamical systems with provable guarantee. This problem captures the core difficulty of underactuated robotics [25]. We demonstrate that neural networks and deep learning can find provably stable controllers in a direct way and tackle the full nonlinearity of the systems, and significantly outperform existing methods based on linear or polynomial approximations such as linear-quadratic regulators (LQR) [17] and sum-of-squares (SOS) and semidefinite programming (SDP) [21]. The results show the promise of neural networks and deep learning in improving the solutions of many challenging problems in nonlinear control.

The prevalent way of stabilizing nonlinear dynamical systems is to linearize the system dynamics around an equilibrium, and formulate LQR problems to minimize deviation from the equilibrium. LQR methods compute a linear feedback control policy, with stability guarantee within a small neighborhood where linear approximation is accurate. However, the dependence on linearization produces extremely conservative systems, and it explains why agile robot locomotion is hard [25]. To control nonlinear systems outside their linearizable regions, we need to rely on Lyapunov methods [13]. Following the intuition that a dynamical system stabilizes when its energy decreases over time, Lyapunov methods construct a scalar field that can force stabilization. These fields are highly nonlinear and the need for function approximations has long been recognized [13]. Many existing approaches rely on polynomial approximations of the dynamics and the search of sum-of-squares polynomials as Lyapunov functions through semidefinite programming (SDP) [21]. A rich theory has been developed around the approach, but in practice the polynomial approximations pose much restriction on the systems and the Lyapunov landscape. Moreover, well-known numerical sensitivity

issues in SDP [18] make it very hard to find solutions that fully satisfy the Lyapunov conditions. In contrast, we exploit the expressive power of neural networks, the convenience of gradient descent for learning, and the completeness of nonlinear constraint solving methods to provide full guarantee of Lyapunov conditions. We show that the combination of these techniques produces control designs that can stabilize various nonlinear systems with verified regions of attraction that are much larger than what can be obtained by existing control methods.

We propose an algorithmic framework for learning control functions and neural network Lyapunov functions for nonlinear systems without any local approximation of their dynamics. The framework consists of a learner and a falsifier. The learner uses stochastic gradient descent to find parameters in both a control function and a neural Lyapunov function, by iteratively minimizing the *Lyapunov risk* which measures the violation of the Lyapunov conditions. The falsifier takes a control function and Lyapunov function from the learner, and searches for *counterexample* state vectors that violate the Lyapunov conditions. The counterexamples are added to the training set for the next iteration of learning, generating an effective curriculum. The falsifier uses delta-complete constraint solving [11], which guarantees that when no violation is found, the Lyapunov conditions are guaranteed to hold for all states in the verified domain. In this framework, the learner and falsifier are given tasks that are difficult in different ways and can not be achieved by the other side. Moreover, we show that the framework provides the flexibility for fine-tuning the control performance by directly enlarging the region of attraction on demand, by adding regulator terms in the learning cost.

We experimented with several challenging nonlinear control problems in robotics, such as drone landing, wheeled vehicle path following, and humanoid robot balancing. We are able to find new control policies that produce certified region of attractions that are significantly larger than what can be established previously. We provide a detailed analysis of the performance comparison between the proposed methods and the LQR/SOS/SDP methods.

**Related Work.** The recent work of Richards *et. al.* [24] has also proposed and shown the effectiveness of using neural networks to learn safety certificates in a Lyapunov framework, but our goals and approaches are different. Richards *et. al.* focus on discrete-time polynomial systems and the use of neural networks to learn the region of attraction of a given controller. The Lyapunov conditions are validated in relaxed forms through sampling. Special design of the neural architecture is required to compensate the lack of complete checking over all states. In comparison, we focus on learning the control and the Lyapunov function together with provable guarantee of stability in larger regions of attraction. Our approach directly handles non-polynomial continuous dynamical systems, does not assume control functions are given other than an initialization, and uses generic feed-forward network representations without manual design. Our approach successfully works on many more nonlinear systems, and find new control functions that enlarge regions of attraction obtainable from standard control methods. Related learning-based approaches for finding Lyapunov functions include [6, 7, 10, 22]. There is strong evidence that linear control functions are all we need for solving highly nonlinear control problems through reinforcement learning as well [20], suggesting convergence of different learning approaches. In the control and robotics community, similar learner-falsifier frameworks have been proposed by [23, 16] without using neural network representations. The common assumption is the Lyapunov functions are high-degree polynomials. In these methods, an explicit control function and Lyapunov function can not be learned together because of the bilinear optimization problems that they generate. Our approach significantly simplifies the algorithms in this direction and has worked reliably on much harder control problems compared to existing methods. Several theoretical results on asymptotic Lyapunov stability [2, 4, 3, 1] show that some very simple dynamical systems do not admit a polynomial Lyapunov function of any degree, despite being globally asymptotically stable. Such results further motivates the use of neural networks as a more suitable function approximator. A large body of work in control uses SOS representations and SDP optimization in the search for Lyapunov functions [14, 21, 9, 15, 19]. However, scalability and numerical sensitivity issues have been the main challenge in practice. As is well known, the number of semidefinite programs from SOS decomposition grows quickly for low degree polynomials [21].

## 2   Preliminaries

We consider the problem of designing control functions to stablize a dynamical system at an equilibrium point. We make extensive use of the following results from Lyapunov stability theory.

**Definition 1** (Controlled Dynamical Systems). An $n$-dimensional controlled dynamical system is

$$\frac{\mathrm{d}x}{\mathrm{d}t} = f_u(x), \quad x(0) = x_0 \tag{1}$$

where $f_u : \mathcal{D} \to \mathbb{R}^n$ is a Lipschitz-continuous vector field, and $\mathcal{D} \subseteq \mathbb{R}^n$ is an open set with $0 \in \mathcal{D}$ that defines the state space of the system. Each $x(t) \in \mathcal{D}$ is a state vector. The feedback control is defined by a continuous function $u : \mathbb{R}^n \to \mathbb{R}^m$, used as a component in the full dynamics $f_u$.

**Definition 2** (Asymptotic Stability). We say that system of (1) is stable at the origin if for any $\varepsilon \in \mathbb{R}^+$, there exists $\delta(\varepsilon) \in \mathbb{R}^+$ such that if $\|x(0)\| < \delta$ then $\|x(t)\| < \varepsilon$ for all $t \geq 0$. The system is asymptotically stable at the origin if it is stable and also $\lim_{t \to \infty} \|x(t)\| = 0$ for all $\|x(0)\| < \delta$.

**Definition 3** (Lie Derivatives). The Lie derivative of a continuously differentiable scalar function $V : \mathcal{D} \to \mathbb{R}$ over a vector field $f_u$ is defined as

$$\nabla_{f_u} V(x) = \sum_{i=1}^{n} \frac{\partial V}{\partial x_i} \frac{\mathrm{d}x_i}{\mathrm{d}t} = \sum_{i=1}^{n} \frac{\partial V}{\partial x_i} [f_u]_i(x)$$

It measures the rate of change of $V$ along the direction of the system dynamics.

**Proposition 1** (Lyapunov Functions for Asymptotic Stability). *Consider a controlled system* (1) *with equilibrium at the origin, i.e., $f_u(0) = 0$. Suppose there exists a continuously differentiable function $V : \mathcal{D} \to \mathbb{R}$ that satisfies the following conditions:*

$$V(0) = 0, \text{ and, } \forall x \in \mathcal{D} \setminus \{0\}, V(x) > 0 \text{ and } \nabla_{f_u} V(x) < 0. \tag{2}$$

*Then, the system is asymptotically stable at the origin and $V$ is called a Lyapunov function.*

Linear-Quadratic Regulators (LQR) is a widely-adpoted optimal control strategy. LQR controllers are guaranteed to work within a small neighborhood around the stationary point where the dynamics can be approximated as linear systems. A detailed description can be found in [17].

## 3    Learning to Stabilize with Neural Lyapunov Functions

We now describe how to learn both a control function and a neural Lyapunov function together, so that the Lyapunov conditions can be rigorously verified to ensure stability of the system. We provide pseudocode of the algorithm in Algorithm 1.

### 3.1    Control and Lyapunov Function Learning

We design the hypothesis class of candidate Lyapunov functions to be multilayered feedforward networks with $\tanh$ activation functions. It is important to note that unlike most other deep learning applications, we can not use non-smooth networks, such as with ReLU activations. This is because we will need to analytically determine whether the Lyapunov conditions hold for these neural networks, which requires the existence of their Lie derivatives.

For a neural network Lyapunov function, its input is any state vector of the system in Definition (1) and the output is a scalar value. We write $\theta$ to denote the parameter vector for a Lyapunov function candidate $V_\theta$. For notational convenience, we write $u$ to denote both the control function and the parameters that define the function. The learning process updates both the $\theta$ and $u$ parameters to improve the likelihood of satisfying the Lyapunov conditions, which we formulate as a cost function named the *Lyapunov risk*. The Lyapunov risk measures the degree of violation of the following Lyapunov conditions, as shown in Proposition (1). First, the value of $V_\theta(x)$ is positive; Second, the value of the Lie derivative $\nabla_{f_u} V_\theta(x)$ is negative; Third, the value of $V_\theta(0)$ is zero. Conceptually, the overall Lyapunov control design problem is about minimizing the minimax cost of the form

$$\inf_{\theta, u} \sup_{x \in \mathcal{D}} \left( \max(0, -V_\theta(x)) + \max(0, \nabla_{f_u} V_\theta(x)) + V_\theta^2(0) \right).$$

The difficulty in control design problems is that the violation of the Lyapunov conditions can not just be estimated, but needs to be fully guaranteed over all states in $\mathcal{D}$. Thus, we need to rely on global search with complete guarantee for the inner maximization part, which we delegate to the falsifier explained in Section 3.2. For the learning step, we define the following Lyapunov risk function.

**Definition 4** (Lyapunov Risk). Consider a candidate Lyapunov function $V_\theta$ for a controlled dynamical system $f_u$ from Definition 1. The Lyapunov risk is a defined by the following function

$$L_\rho(\theta, u) = \mathbb{E}_{x \sim \rho(\mathcal{D})}\Big( \max(0, -V_\theta(x)) + \max(0, \nabla_{f_u} V_\theta(x)) + V_\theta^2(0) \Big), \qquad (3)$$

where $x$ is a random variable over the state space of the system with a distribution $\rho$. In practice, we work with the Monte Carlo estimate named the *empirical Lyapunov risk* by drawing samples:

$$L_{N,\rho}(\theta, u) = \frac{1}{N} \sum_{i=1}^{N} \Big( \max(-V_\theta(x_i), 0) + \max(0, \nabla_{f_u} V_\theta(x_i)) \Big) + V_\theta^2(0), \qquad (4)$$

where $x_i, 1 \le i \le N$ are samples of the state vectors sampled according to $\rho(\mathcal{D})$.

It is clear that the empirical Lyapunov risk is an unbiased estimator of the Lyapunov risk function. It is clear that $L_{N,\rho}$ is an unbiased estimator of $L_\rho$.

Note that $L_\rho$ is positive semidefinite, and any $(\theta, u)$ that corresponds to a true Lyapunov function satisfies $L(\theta, u)$=0. Thus, Lyapunov functions define global minimizers of the Lyapunov risk function.

**Proposition 2.** *Let $V_{\theta_o}$ be a Lyapunov function for dynamical system $f_{u_o}$ where $u_o$ is the control parameters. Then $(\theta_o, u_o)$ is a global minimizer for $L_\rho$ and $L_\rho(\theta_o, u_o) = 0$.*

Note that both $V_\theta$ and $f_u$ are highly nonlinear (even though $u$ is almost always linear in practice), and thus $L(\theta, u)$ generates a highly complex landscape. Surprisingly, multilayer feedforward $\tanh$ networks and stochastic gradient descent can quickly produce generalizable Lyapunov functions with nice geometric properties, as we report in detail in the experiments. In Figure 1 (b), we show an example of how the Lyapunov risk is minimized over iterations on the inverted pendulum example.

**Initialization and improvement of control performance over LQR.** Because of the local nature of stochastic gradient descent, it is hard to learn good control functions through random initialization of control parameters. Instead, the parameters $u$ in the control function are initialized to the LQR solution, obtained for the linearized dynamics in a small neighborhood around the stationary point. On the other hand, the initialization of the neural network Lyapunov functions can be completely random. We observe that the final learned controller often delivers significantly better control solutions than the initalization from LQR. Figure 1(a) shows how the learned control reduces oscillation of the system behavior in the humanoid robot balancing example and achieve more stable control.

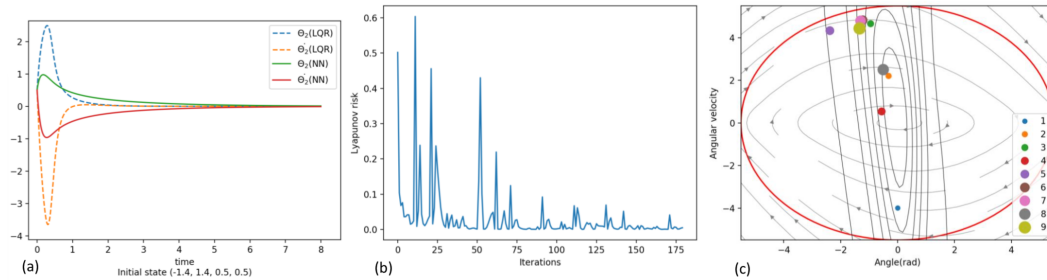

Figure 1: (a) Comparison between LQR and deep-learned controllers for humanoid balancing. (b) The Lyapunov risk decreases quickly over iterations. (c) Counterexamples returned by falsifiers from several epochs, which quickly guides the learner to focus on sepcial regions in the space.

## 3.2 Falsification and Counterexample Generation

For each control and Lyapunov function pair $(V_\theta, u)$ that the learner obtains, the falsifier's task is to find states that violate the Lyapunov conditions in Proposition 1. We formulate the *negations* of the Lyapunov conditions as a nonlinear constraint solving problem over real numbers. These *falsification constraints* are defined as follows.

**Definition 5** (Lyapunov Falsification Constraints). Let $V$ be a candidate Lyapunov function for a dynamical system defined by $f_u$ defined in state space $\mathcal{D}$. Let $\varepsilon \in \mathbb{Q}^+$ be a small constant parameter that bounds the tolerable numerical error. The Lyapunov falsification constraint is the following first-order logic formula over real numbers:

$$\Phi_\varepsilon(x) := \left( \sum_{i=1}^n x_i^2 \geq \varepsilon \right) \wedge \left( V(x) \leq 0 \vee \nabla_{f_u} V(x) \geq 0 \right)$$

where $x$ is bounded in the state space $\mathcal{D}$ of the system. The numerical error parameter $\varepsilon$ is explicitly introduced for controlling numerical sensitivity near the origin. Here $\varepsilon$ is orders of magnitude smaller than the range of the state variables, i.e., $\sqrt{\varepsilon} \ll \min(1, ||\mathcal{D}||_2)$.

**Remark 1.** The numerical error parameter $\varepsilon$ allows us to avoid pathological problems in numerical algorithms such as arithmetic underflow. Values inside this tiny ball correspond to disturbances that are physically insignificant. This parameter is important for eliminating from our framework the numerical sensitivity issues commonly observed in SOS/SDP methods. Also note the $\varepsilon$-ball does not affect properties of the Lyapunov level sets and regions of attraction outside it (i.e., $\mathcal{D} \setminus B_\varepsilon$).

The falsifier computes solutions of the falsification constraint $\Phi_\varepsilon(x)$. Solving the constraints requires global minimization of a highly nonconvex functions (involving Lie derivatives of the neural network Lyapunov function), and it is a computationally expensive task (NP-hard). We rely on recent progress in nonlinear constraint solving in SMT solvers such as dReal [11], which has been used for similar control design problems [16] that do not involve neural networks.

**Example 1.** *Consider a candidate Lyapunov function $V(x) = \tanh(a_1 x_1 + a_2 x_2 + b)$ and dynamics $\dot{x}_1 = -x_2^2$ and $\dot{x}_2 = \sin(x_1)$. The falsification constraint is of the following form*

$$\Phi_\varepsilon(x) := (x_1^2 + x_2^2) \geq \varepsilon \wedge (\tanh(a_1 x_1 + a_2 x_2 + b) \leq 0 \vee a_1(1 - \tanh^2(a_1 x_1 + a_2 x_2 + b))(-x_2^2)$$
$$+ a_2(1 - \tanh^2(a_1 x_1 + a_2 x_2 + b)) \sin(x_1) \geq 0))$$

*which is a nonlinear non-polynomial disjunctive constraint system. The actual examples used in our experiments all use larger two-layer $\tanh$ networks and much more complex dynamics.*

To completely certify the Lyapunov conditions, the constraint solving step for $\Phi_\varepsilon(x)$ can never fail to report solutions if there is any. This requirement is rigorously proved for algorithms in SMT solvers such as dReal [12], as a property called delta-completeness [11].

**Definition 6** (Delta-Complete Algorithms). Let $C$ be a class of quantifier-free first-order constraints. Let $\delta \in \mathbb{Q}^+$ be a fixed constant. We say an algorithm $\mathcal{A}$ is $\delta$-complete for $C$, if for any $\varphi(x) \in C$, $\mathcal{A}$ always returns one of the following answers correctly: $\varphi$ does not have a solution (unsatisfiable), or there is a solution $x = a$ that satisfies $\varphi^\delta(a)$. Here, $\varphi^\delta$ is defined as a small syntactic variation of the original constraint (precise definitions are in [11]).

In other words, if a delta-complete algorithm concludes that a formula $\Phi_\varepsilon(x)$ is unsatisfiable, then it is guaranteed to not have any solution. In our context, this is exactly what we need for ensuring that the Lyapunov condition holds over all state vectors. If $\Phi_\varepsilon(x)$ is determined to be $\delta$-satisfiable, we obtain counterexamples that are added to the training set for the learner. Note that the counterexamples are simply state vectors without labels, and their Lyapunov risk will be determined by the learner, not the falsifier. Thus, although it is possible to have spurious counterexamples due to the $\delta$ error, they are used as extra samples and do not harm correctness of the end result. In all, when delta-complete algorithms in dReal return that the falsification constraints are unsatisfiable, we conclude that the Lyapunov conditions are satisfied by the candidate Lyapunov and control functions. Figure 1(c) shows a sequence of counterexamples found by the falsifier to improve the learned results.

**Remark 2.** When solving $\Phi_\varepsilon(x)$ with $\delta$-complete constraint solving algorithms, we use $\delta \ll \varepsilon$ to reduce the number of spurious counterexamples. Following delta-completeness, the choice of $\delta$ does not affect the guarantee for the validation of the Lyapunov conditions.

### 3.3 Tuning Region of Attraction

An important feature of the proposed learning framework is that we can adjust the cost functions to learn control and Lyapunov functions favoring various additional properties. In fact, the most practically important performance metric for a stabilizing controller is its region of attraction (ROA).

An ROA defines a forward invariant set that is guaranteed to contain all possible trajectories of the system, and thus can conclusively prove safety properties. Note that the Lyapunov conditions themselves do not directly ensure safety, because the system can deviate arbitrarily far before coming back to the stable equilibrium. Formally, the ROA of an asymptotically stable system is defined as:

**Definition 7** (Region of Attraction). Let $f_u$ define a system asymptotically stable at the origin with Lyapunov function $V$ for domain $\mathcal{D}$. A region of attraction $R$ is a subset of $\mathcal{D}$ that contains the origin and guarantees that the system never leaves $R$. Any level set of $V$ completely contained in $\mathcal{D}$ defines an ROA. That is, for $\beta > 0$, if $R_\beta = \{V(x) \le \beta\} \subseteq \mathcal{D}$, then $R_\beta$ is an ROA for the system.

To maximize the ROA produced by a pair of Lyapunov function and control function, we add a cost term to the Lyapunov risk that regulates how quickly the Lyapunov function value increases with respect to the radius of the level sets, by using $L_{N,p}(\theta, u) + \frac{1}{N}\sum_{i=1}^{N} \|x_i\|_2 - \alpha V_\theta(x_i)$ following Definition 4. Here $\alpha$ is tunable parameter. We observe that the regulator can have major effect on the performance of the learned control functions. Figure 2 illustrates such an example, showing how different control functions are obtained by regulating the Lyapunov risk to achieve larger ROA.

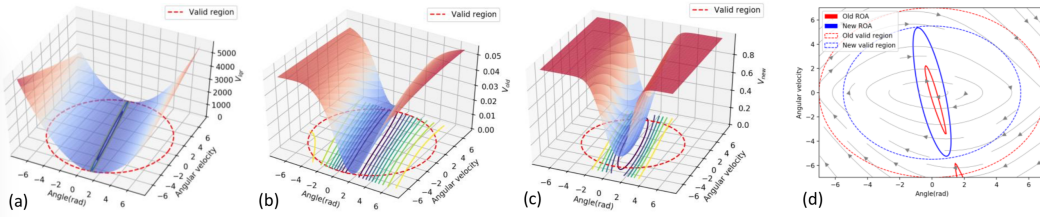

Figure 2: (a) Lyapunov function found by the initial LQR controller. (b) Lyapunov function found by learning without tuning the ROA. (c) Lyapunov function found by learning after adding the ROA tuning term. (d) Comparison of ROA for the different Lyapunov functions.

---

**Algorithm 1** Neural Lyapunov Control

1: **function** LEARNING($X, f, q^{lqr}$)
2:  Set learning rate $(0.01)$, input dimension (# of state variables), output dimension $(1)$
3:  Initialize feedback controller $u$ to LQR solution $q^{lqr}$
4:  **Repeat:**
5:    $V_\theta(x), u(x) \leftarrow \text{NN}_{\theta,u}(x)$           ▷ Forward pass of neural network
6:    $\nabla_{f_u} V_\theta(x) \leftarrow \sum_{i=1}^{D_{in}} \frac{\partial V}{\partial x_i}[f_u]_i(x)$
7:    Compute Lyapunov risk $L(\theta, u)$
8:    $\theta \leftarrow \theta + \alpha \nabla_\theta L(\theta, u)$
9:    $u \leftarrow u + \alpha \nabla_u L(\theta, u)$           ▷ Update weights using SGD
10:  **Until** convergence
11:  **return** $V_\theta, u$
12: **end function**
13: **function** FALSIFICATION($f, u, V_\theta, \varepsilon, \delta$)
14:  Encode conditions in Definition 5
15:  Using SMT solver with $\delta$ to verify the conditions
16:  **return** satisfiability
17: **end function**
18: **function** MAIN( )
19:  **Input:** dynamical system $(f)$, parameters of LQR $(q^{lqr})$, radius $(\varepsilon)$, precision $(\delta)$ and an initial set of randomly sampled states in $D$
20:  **while** Satisfiable **do**
21:    Add counterexamples to $X$
22:    $V_\theta, u \leftarrow$ LEARNING($X, f, q^{lqr}$)
23:    CE$\leftarrow$ FALSIFICATION($f, u, V_\theta, \varepsilon, \delta$)
24:  **end while**
25: **end function**

# 4  Experiments

We demonstrate that the proposed methods find provably stable control and Lyapunov functions on various nonlinear robot control problems. In all the examples, we use a learning rate of $0.01$ for the learner, an $\varepsilon$ value of $0.25$ and $\delta$ value of $0.01$ for the falsifier, and re-verify the result with smaller $\varepsilon$ in Table 1. We emphasize that the choices of these parameters do not affect the stability guarantees on the final design of the control and Lyapunov functions. We show that the region of attraction is enlarged by 300% to 600% compared to LQR results in these examples. Full details of the results and system dynamics are provided in the Appendix. Note that for the Caltech ducted fan and humanoid balancing examples, we numerically relaxed the conditions slightly when the learning has converged, so that the SMT solver dReal does not run into numerical issues. More details on the effect of such relaxation can be found in the paper website [8].

| Benchmarks | Learning time | falsification time | # samples | # iterations | $\varepsilon$ |
|---|---|---|---|---|---|
| Inverted Pendulum | 25.5 | 0.6 | 500 | 430 | 0.04 |
| Path Following | 36.3 | 0.2 | 500 | 610 | 0.01 |
| Caltech Ducted Fan | 1455.16 | 50.84 | 1000 | 3000 | 0.01 |
| Humanoid Balancing | 6000 | 458.27 | 1000 | 4000 | 0.01 |

Table 1: Runtime statistics of the full procedures on four nonlinear control examples.

**Inverted pendulum.**  The inverted pendulum is a standard nonlinear control problem for testing different control methods. This system has two state variables, the angular position $\theta$, angular velocity $\dot{\theta}$ and one control input $u$. Our learning procedure finds a neural Lyapunov function that is proved to be valid within the domain $\|x\|_2 \leq 6$. In contrast, the ROA found by SOS/SDP techniques is an ellipse with large diameter of $1.75$ and short diameter of $1.2$. Using LQR control on the linearized dynamics, we obtain an ellipse with large diameter of $6$ and short diameter of $0.1$. We observe that among all the examples in our experiments, this is the only one where the SOS Lyapunov function has passed the complete check by the constraint solver, so that we can compare to it. The Lyapunov function obtained by LQR gives a larger ROA if we ignore the linearization error. The different regions of attractions are shown in Figure 3. These values are consistent with the approximate maximum region of attraction reported in [24]. In particular, Figure 3 (c) shows that the SOS function does not define a big enough ROA, as many trajectories escape its region.

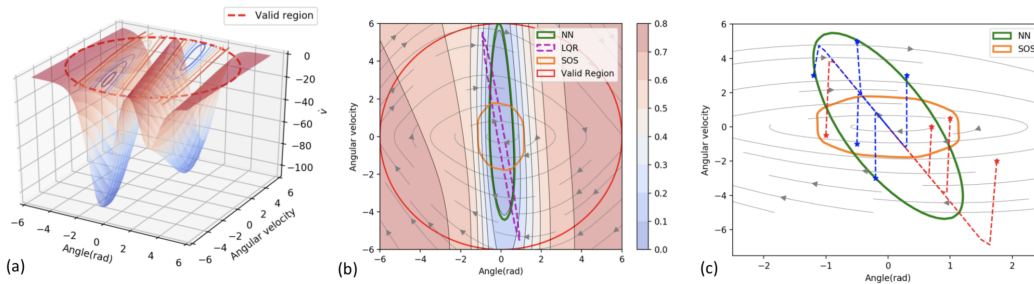

Figure 3: Results of Lyapunov functions for inverted pendulum. (a) Lie derivative of learned Lyapunov function over valid region. Its value is negative over the valid region, satisfying the Lyapunov conditions. (b) ROA estimated by different Lyapunov functions. Our method enlarges the ROA from LQR three times. (c) Validation of ROAs. Stars represent initial states. It shows trajectories start near border of the ROA defined by the learned neural Lyapunov function are safely bounded within the green region. On the contrary, many trajectories (red) starting inside the SOS region can escape, and thus the region fails to satisfy the ROA properties.

**Caltech ducted fan in hover mode.** The system describes the motion of a landing aircraft in hover mode with two forces $u_1$ and $u_2$. The state variables $x$, $y$, $\theta$ denote the position and orientation of the centre of the fan. There are six state variables $[x, y, \theta, \dot{x}, \dot{y}, \dot{\theta}]$. The dynamics, neural Lyapunov

function with two layers of $\texttt{tanh}$ activation functions, and the control policy are given in the Appendix. In Figure $4(a)$, we show that the ROA is significantly larger than what can be obtained from LQR.

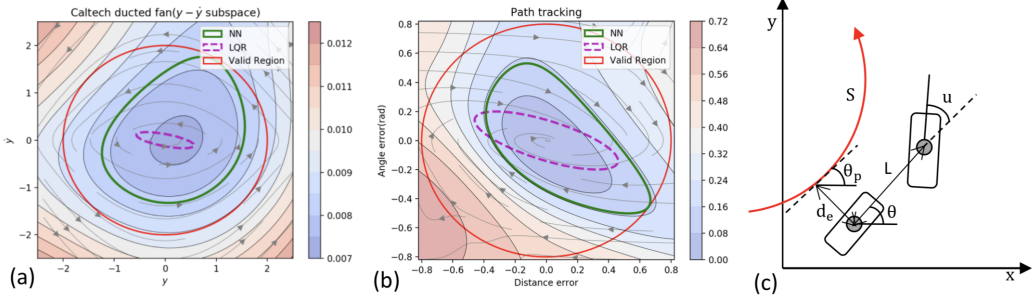

Figure 4: (a) Comparison of ROAs for Caltech ducted fan. (b) Comparison of ROAs for path following. (c) Schematic diagram of wheeled vehicle to show the nonlinear dynamics.

**Wheeled vehicle path following.** We consider the path tracking control using kinematic bicycle model (see Figure $4(c)$). We take the angle error $\theta_e$ and the distance error $d_e$ as state variables. Assume a target path is a unit circle, then we obtain the Lyapunov function within $\|x\|_2 \leq 0.8$.

**Humanoid balancing.** The task of balancing a humanoid robot can be modelled as maintaining an $n$-link pendulum a vertical posture. The $n$-link pendulum system has $n$ control inputs and $2n$ state variables $[\theta_1, \theta_2, \ldots, \theta_n, \dot{\theta}_1, \dot{\theta}_2, \ldots, \dot{\theta}_n]$, representing the $n$ link angles and $n$ angle velocities. Each link has mass $m_i$ and length $\ell_i$, and the moments of inertia $I_i$ are computed from the link pivots, where $i = 1, 2, \ldots, n$. We find a neural Lyapunov function for the 3-link pendulum system within $\|x\|_2 \leq 0.5$. In Figure 5, we show the shape of the neural Lyapunov functions on two of the dimensions, and the ROA that the control design achieves. We also provide a video of the control on the 3-link model.

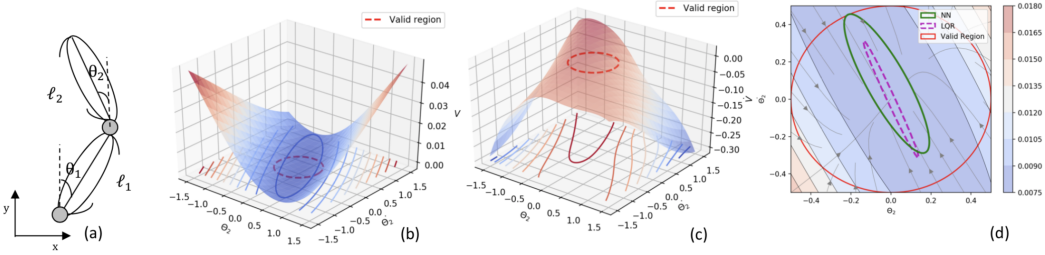

Figure 5: Results of humanoid balance. (a) Schematic diagram. (b) Learned Lyapunov function. (c) Lie derivative of Lyapunov function. (d) Comparison of the region of attraction.

## 5   Conclusion

We proposed new methods to learn control policies and neural network Lyapunov functions for highly nonlinear systems with provable guarantee of stability. The approach significantly simplifies the process of nonlinear control design, provides end-to-end provable correctness guarantee, and can obtain much larger regions of attraction compared to existing control methods. We show experiments on challenging nonlinear problems central to various robotics problems. The proposed methods demonstrate clear advantage over existing methods. We envision that neural networks and deep learning will provide immediate solutions to many core problems in robot control design.

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
