[Supplementary Material]

Appendices to the Neural Lyapunov Control paper.

## Appendix A: Further Details on Examples in Section 3

**Example 1.** *In the double inverted pendulum model for humanoid robot balancing (Section 4), the learned control functions are*

$$u_1(x) = -39.51138\theta_1 - 20.1297\theta_2 - 21.4826\dot{\theta}_1 - 10.0516\dot{\theta}_2$$
$$u_2(x) = -18.2287\theta_1 - 19.143\theta_2 - 9.62847\dot{\theta}_1 - 10.480879\dot{\theta}_2,$$

*and the initial solution from LQR solution is*

$$u_1^*(x) = -49.524906\theta_1 - 16.9531480\theta_2 - 17.04338\dot{\theta}_1 - 10.051599\dot{\theta}_2$$
$$u_2^*(x) = -16.119429\theta_1 - 19.143\theta_2 - 9.2905\dot{\theta}_1 - 6.6695\dot{\theta}_2.$$

*Figure* $5(d)$ *shows that the learned results give much better performance. In fact, the differences contribute to the improvement in region of attraction by three times.*

## Appendix B: Experiments

We demonstrate that the proposed methods can find Lyapunov functions and control functions on various nonlinear robotics control problems.

**Inverted pendulum.** The inverted pendulum is one of the most standard nonlinear control problem for testing different control methods. This system has two state variables $\theta$, $\dot{\theta}$ and one control input $u$. $\theta$ and $\dot{\theta}$ represent the angular position from the inverted position and angular velocity. The system dynamics can be described as

$$\ddot{\theta} = \frac{mg\ell \sin(\theta) + u - 0.1\dot{\theta}}{m\ell^2} \tag{1}$$

Using constants $g = 9.81$, $m = 0.15$ and $\ell = 0.5$, our learning procedure finds the following neural Lyapunov function: $V = \tanh(W_2 \tanh(W_1 x + B_1) + B_2)$, where $x = [\theta \ \dot{\theta}]^T$ and

$$W_1 = \begin{bmatrix} -1.1751 & 0.0265 & 0.0439 & -0.5518 & -0.0067 & 0.4446 \\ 0.0288 & -0.0007 & -0.0030 & -0.0348 & -0.0067 & 0.2599 \end{bmatrix}^T,$$

$$W_2 = \begin{bmatrix} 0.6047 & -0.6942 & -1.1177 & -1.1330 & 0.7800 & -0.2621 \end{bmatrix},$$

$$B_1 = \begin{bmatrix} -1.2251 & -0.8158 & -0.5308 & 0.8925 & 0.9339 & 1.0895 \end{bmatrix} \text{ and } B_2 = \begin{bmatrix} 0.1592 \end{bmatrix},$$

and the linear control function, $u = -23.1717\theta - 6.7996\dot{\theta}$.

**Caltech ducted fan in hover mode.** This dynamics describes the motion of a landing aircraft in a hover mode with two forces $u_1$ and $u_2$. Let $x$, $y$, $\theta$ denote the position and orientation of the centre of the fan, then the dynamics with six state variables $[x, y, \theta, \dot{x}, \dot{y}, \dot{\theta}]$ of motion can be written as follows:

$$\ddot{x} = \frac{-d\dot{x} + u_1 \cos(\theta) - u_2 \sin(\theta)}{m},$$
$$\ddot{y} = \frac{-d\dot{y} + u_1 \sin(\theta) - u_2 \cos(\theta) - mg}{m}, \tag{2}$$
$$\ddot{\theta} = \frac{ru_1}{I},$$

where $g = 0.28$, $m = 11.2$, $I = 0.0462$, $r = 0.156$ and $d = 0.1$. We finds a neural Lyapunov function: $V = \tanh(W_2 \tanh(W_1 x + B_1) + B_2)$, where $x = [x \ y \ \theta \ \dot{x} \ \dot{y} \ \dot{\theta}]^T$ and

$$W_1 = \begin{bmatrix} 0.0314 & 0.0190 & -0.1893 & 0.2532 & 0.0177 & -0.0890 \\ -0.0397 & 0.0242 & 0.1094 & -0.1346 & 0.0186 & 0.1177 \\ -0.1221 & 0.0584 & 0.1417 & -0.0897 & -0.0658 & 0.0060 \\ -0.0853 & -0.0682 & -0.0680 & 0.1741 & 0.2397 & 0.0061 \\ 0.0847 & 0.0065 & 0.0952 & -0.1782 & 0.3689 & 0.0006 \\ -0.1239 & 0.2481 & -0.0991 & 0.2475 & -0.0408 & 0.0017 \end{bmatrix}$$

,
$$W_2 = \begin{bmatrix} 0.0563 & 0.0368 & 0.0218 & -0.0158 & -0.0093 & -0.0186 \end{bmatrix},$$
$$B_1 = \begin{bmatrix} -0.6099 & -0.5518 & 0.1146 & 0.1873 & 0.2220 & 0.4308 \end{bmatrix} \text{ and } B_2 = \begin{bmatrix} 0.0666 \end{bmatrix}$$

and two neural controllers:

$$u_1 = 0.5000x + 0.000002y - 2.1339\theta + 2.7899\dot{x} - 0.00000003\dot{y} - 1.3992\dot{\theta}$$

$$u_2 = 0.000001x - 1.0000y - 0.000003\theta - 0.000003\dot{x} - 5.0407\dot{y} - 0.000001\dot{\theta}$$

**Humanoid balance.** The task of balancing humanoid robot can simplify to maintain $n$-link pendulum a vertical posture. The $n$-link pendulum system has $n$ control inputs and $2n$ state variables $[\theta_1, \theta_2, \ldots, \theta_n, \dot{\theta}_1, \dot{\theta}_2, \ldots, \dot{\theta}_n]$, where represent the $n$ link angle and the $n$ angle velocity. Let each link has mass $m_i$ and length $\ell_i$, and the moments of inertia $I_i$ are computed from the link pivots, where $i = 1, 2, \ldots, n$, then the dynamics has the form:

$$M(\theta)\ddot{\theta} + C(\theta, \dot{\theta})\dot{\theta} + \tau(\theta) = Bu, \tag{3}$$

where

$$\theta = [\theta_1, \theta_2, \ldots, \theta_n]^{\mathrm{T}} \in \mathbb{R}^n, u \in \mathbb{R}^n$$

$$M(\theta) = [a_{ij}\cos(\theta_j - \theta_i)], M(\theta) \in \mathbb{R}^{n \times n}$$

$$C(\theta, \dot{\theta}) = \left[-a_{ij}\dot{\theta}_j\sin(\theta_j - \theta_i)\right], C(\theta, \dot{\theta}) \in \mathbb{R}^{n \times n},$$

$$\tau(\theta) = [-b_i\sin\theta_i], G(\theta) \in \mathbb{R}^n,$$

$$B = [1, 1, \ldots, 1]^{\mathrm{T}}$$

$$\begin{cases} a_{ii} = I_i + m_i\ell_{ci}^2 + \ell_i^2\sum_{k=i+1}^n m_k, 1 \le i \le n \\ a_{ij} = a_{ji} = m_j\ell_i\ell_{cj} + \ell_i\ell_j\sum_{k=j+1}^n m_k, 1 \le i < j \le n \end{cases}$$

$$b_i = \left(m_i\ell_{ci} + \ell_i\sum_{k=i+1}^n m_k\right)g, 1 \le i \le n,$$

For the 2-link pendulum system our approach can find the following neural Lyapunov function that is valid within domain $\mathcal{D} : \|x\|_2 \le 0.5$ under precision $\delta = 0.01$: $V = \tanh(W_2\tanh(W_1x + B_1) + B_2)$, where $x = [\theta_1\ \theta_2\ \dot{\theta}_1\ \dot{\theta}_2]^T$

$$W_1 = \begin{bmatrix} -0.3578 & -0.2339 & -0.5153 & -0.2648 \\ 0.4244 & 0.3886 & 0.1041 & 0.0195 \\ -0.4218 & -0.4314 & -0.4371 & -0.2353 \\ -0.0042 & -0.0020 & -0.0013 & -0.0077 \end{bmatrix},$$

$$W_2 = \begin{bmatrix} 0.1670 & -0.1353 & -0.2582 & 0.5208 \end{bmatrix},$$

$$B_1 = \begin{bmatrix} -0.4547 & 0.0263 & 0.6899 & 0.7721 \end{bmatrix} \text{ and } B_2 = \begin{bmatrix} 0.7633 \end{bmatrix},$$

and two neural controllers are

$$u_1 = -49.5249\theta_1 - 20.0854\theta_2 - 21.4826\dot{\theta}_1 - 10.0516\dot{\theta}_2$$

$$u_2 = -18.2287\theta_1 - 19.143\theta_2 - 9.2905\dot{\theta}_1 - 6.6695\dot{\theta}_2$$

Also, the learning procedure finds a neural Lyapunov function for the 3-link pendulum system on valid domain $\mathcal{D} : \|x\|_2 \le 0.5$ under precision $\delta = 0.01$. The neural Lyapunov function: $V = \tanh(W_2\tanh(W_1x + B_1) + B_2)$, where $x = [\theta_1\ \theta_2\ \theta_3\ \dot{\theta}_1\ \dot{\theta}_2\ \dot{\theta}_3]^T$

$$W_1 = \begin{bmatrix} -0.1919 & 0.1715 & -0.0481 & 0.0707 & 0.1923 & 0.0548 \\ 0.0943 & 0.0112 & 0.0027 & 0.0102 & -0.0005 & 0.0002 \\ 0.0942 & -0.2393 & 0.0932 & -0.0692 & -0.1582 & -0.0221 \\ -0.1136 & -0.1927 & -0.0753 & -0.0407 & -0.1289 & 0.0246 \\ -0.1645 & 0.2017 & 0.0412 & -0.1091 & -0.1892 & -0.1396 \\ 0.0868 & 0.0103 & 0.0030 & 0.0094 & -0.0002 & -0.0007 \end{bmatrix},$$

$$W_2 = \begin{bmatrix} 0.0017 & 0.4299 & 0.0023 & -0.0021 & 0.0002 & -0.5047 \end{bmatrix},$$

$$B_1 = \begin{bmatrix} -0.5246 & -0.3993 & -0.3698 & 0.1214 & 0.2343 & 0.4633 \end{bmatrix} \text{ and } B_2 = \begin{bmatrix} 0.3918 \end{bmatrix},$$

and three neural controllers are

$$u_1 = -101.7856\theta_1 - 8.9265\theta_2 - 3.467\theta_3 - 28.5081\dot{\theta}_1 - 14.0951\dot{\theta}_2 - 7.3643\dot{\theta}_3$$

$$u_2 = 15.8736\theta_1 - 62.5769\theta_2 - 4.0104\theta_3 - 7.8591\dot{\theta}_1 - 12.6341\dot{\theta}_2 - 7.3690\dot{\theta}_3$$

$$u_3 = 5.1672\theta_1 + 7.2750\theta_2 - 42.4820\theta_3 - 2.6997\dot{\theta}_1 - 4.9186\dot{\theta}_2 - 11.8446\dot{\theta}_3$$

**Wheeled vehicle path following.** We consider the path tracking control using kinematic bicycle model from (see Figure $4(c)$). We take the angle error $\theta_e$ and the distance error $d_e$ as state variables, which $\theta_e = \theta - \theta_p$, then the system can be written as the form:

$$
\begin{aligned}
\dot{s} &= \frac{v \cos(\theta_e)}{1 - \dot{d}_e \kappa(s)}, \\
\dot{d}_e &= v \sin(\theta_e), \\
\dot{\theta}_e &= \frac{v \tan(u)}{L} - \frac{v \kappa(s) \cos(\theta_e)}{1 - \dot{d}_e \kappa(s)}.
\end{aligned}
\tag{4}
$$

Assume a target path is a unit circle, then we obtain the following Lyapunov function on for $\|x\|_2 \leq 0.8$, $V = \tanh(W_2 \tanh(W_1 x + B_1) + B_2)$, where $x = [d_e\ \theta_e]^T$ and

$$
W_1 = \begin{bmatrix} -2.5250 & -0.4774 & -0.5239 & -0.0232 & -0.0627 & 1.3562 \\ -0.1841 & -0.6964 & -0.5862 & -0.5032 & -0.5620 & 2.5184 \end{bmatrix}^T,
$$

$$
W_2 = \begin{bmatrix} 0.6251 & -1.0490 & -1.0708 & 0.4644 & 0.7019 & -1.1287 \end{bmatrix},
$$

$$
B_1 = \begin{bmatrix} -1.2776 & -0.4641 & -0.3699 & 0.9194 & 0.9758 & 1.3282 \end{bmatrix} \text{ and } B_2 = \begin{bmatrix} 0.0997 \end{bmatrix}
$$

and the neural controller is $u = -0.8471 d_e - 1.6414 \theta_e$.

## Appendix C: More Details on Related Work

Compared to the control-theoretic approaches, neural Lyapunov control provides a much simpler design process, relying purely on gradient-based methods for the learning. The saving is similar to the reduction of feature engineering and specific optimization methods in other areas of AI. The recent work of Richards *et. al.* [16] has also proposed and shown the effectiveness of using neural networks to learn safety certificates in a Lyapunov framework, but our goals and approaches are different. Richards *et. al.* focus on discrete-time polynomial systems and the use of neural networks to learn the region of attraction of a given controller. The Lyapunov conditions are validated in relaxed forms through sampling. Special design of the neural architecture is required to compensate the lack of complete checking over all states. In comparison, we focus on learning the control and the Lyapunov function together with provable guarantee of stability in larger regions of attraction. Our approach directly handles non-polynomial continuous dynamical systems, does not assume control functions are given other than an initialization, and uses generic feed-forward network representations without manual design. Our approach successfully works on many more nonlinear systems, and find new control functions that enlarge regions of attraction obtainable from standard control methods. Related learning-based approaches for finding Lyapunov functions include [4, 5, 7, 14]. There is strong evidence that linear control functions are all we need for solving nonlinear control problems in reinforcement learning as well [12], suggesting convergence of different learning approaches.

Similar to our approach, in Revanbakhsh *et. al.* [15] there is a candidate function which is falsified by a verifier, and a learner who updates the function based on counterexamples provided by the verifier. However, there important differences in our approaches. Most importantly, we directly search for a controller as an explicit function of states, which is important for reliability, whereas Revanbakhsh *et. al.* look for a control Lyapunov function first, and then use results from [17] to construct a non-linear controller that is *not* necessarily continuous at the origin. Another difference is that Revanbakhsh *et. al.* use semi-definite programming (SDP) for the verification to improve efficiency of their algorithm, whereas we use $\delta$-complete decision procedures. The methods do not support non-polynomial dynamics and use Taylor expansion to represent trigonometric functions using polynomials, unable to provide complete guarantee about the effect of the learned control over the original system. Using SDP for the verification involves relaxations of constraints it is possible that verifier in [15] rejects a perfectly fine candidate.

Kapinski *et. al.* [10] use simulation to improve efficiency of the search for Lyapunov functions. Similar to our work, they do not rely on local approximation of the dynamics, but have to assume the Lyapunov function is represented as a low-degree polynomial. They only consider switching systems with no control input, and look for Lyapunov functions of the form $z^T P z$, where $z$ is a vector of $m$ monomials and $P$ is a symmetric matrix with elements in $R$. Whenever their algorithm finds a candidate Lyapunov function, it uses SMT solvers like Z3 or dReal to verify conditions of a Lyapunov

function. If the verification fails, a counterexample is returned which is used to update matrix $P$ in the candidate $z^T P z$. Since the verification is very expensive, authors first use simulations to construct a set of linear constraints that a Lyapunov function in the form of $z^T P z$ must *not* satisfy. As long as these constraints are satisfiable, counterexample executions are used to update them. Verification is only involved when those constraints are proven to be unsatisfiable.

Majumdar *et. al.* [11], use sum of squares (SOS) and semidefinite programs to guarantee validity of their solutions to motion planning problems for non-linear systems with uncertainty. Given a source and destination in a bounded domain of interest, they find a *funnel* and control inputs that guarantees as long as uncertainties and disturbances in the real world are considered in the over-approximations of the model, robot can stay inside the funnel (and hence avoid unsafe states) and reaches the goal.

Ahmadi *et. al.* [2] prove that dynamical system $\dot{x} = -x + xy, \dot{y} = -y$ does not admit a polynomial Lyapunov function of any degree, despite being globally asymptotically stable. It is common to use sum of squares (SOS) optimization in the search for Lyapunov functions [8, 13, 6, 9, 11]. However, scalability is arguably the most outstanding challenge for this method; if a polynomial over $n$ variables has degree $2d$ then the size of the corresponding semidefinite program from SOS decomposition will be roughly $n^d$ which can grow quickly even for low degree polynomials. An interesting relaxation to Lyapunov function comes from the fact that in order to eventually reach and stay at zero, a function does not have to be monotonic. This allows Lyapunov functions that could rise but always compensate for that rise [1, 3] motivating for more expressive nonlinear function approximators.