[Reviews · NeurIPS 2019]

Reviewer 1



The topic of this paper is highly relevant, since stability guarantees are often sought after for learned policies. While I'm generally excited about the approach, the paper does not address aspects that might make the technique applicable to domains with non-smooth (e.g. legged locomotion) or unknown dynamics (e.g. a real robot or even a physics simulator). The condition in Eq. 1 is a strong one and I understand the theoretical need for it. However, I'm wondering if there's any value to the proposed technique in case this condition is not met? Will the inclusion of the Lyapunov risk as a term in the cost function yield feedback controllers that are more robust in practice, even for non-smooth systems? A few more detailed comments: - You mention that Relu based networks can't be supported from a theoretical point of view. Does this matter in practice? - Have you tried your approach on any more complex systems or using a learned (fitted) dynamics model rather than a given one? - Please provide learning curves for the various experiments. As a reader, I have no idea how complicated it is to learn the stability guarantees as compared to just optimizing a normal feedback controller. How often does it fail? Is failure to find a stable controller more often the case for larger (more DoF) system. Originality: Good. This paper combines Lyapunov theory and learning based control. The basic algorithm (without the SMT solver) appears to be relatively straightforward to add to an existing setup (given it has differentiable dynamics). Quality: OK. The experiments seem difficult to reproduce with the information provided. Clarity: OK. The writing itself is of high quality. However, the paper could be better organized. As a reader, I'm missing a concise overview of how the algorithm works. Significance: High. If it's possible to obtain stability guarantees as part of the learning process for non-trivial tasks, then that's a significant step forward. This paper appears to be a step in the right direction.

Reviewer 2



The core of the method is the interaction between the learner and the falsifier. The learner minimizes the Lyapunov risk, defined as the maximum violation of the Lyapunov stability condition over the state space. The falsifier solves a non-linear constraints feasibility problem where it tries to find a state that violates the Lyapunov stability criterion given the current Lyapunov function and controller. Due to the delta-completeness of the falsifier, if no such example can be found, the controller is guaranteed to stabilize the system. The originality of the work is not clearly defined. In the related work section, the authors mention other works that use NN to learn Lyapunov functions as well as other works that use similar learner-falsifier frameworks which use the same non-linear constraint solver to learn controllers for non-linear systems. As such, it is not easy to assess the novelty of the paper. The clarity of the work could be improved. The authors use a good amount of space to report things that do not really give an insight into the paper while they introduce central concepts in just a few lines. For example, the authors unroll the definition of Lyapunov falsification constraints in example 1 for a simple NN and a simple controller and they report the numerical values of the entire matrices learned by their method in section 4. In my opinion, these do not help to understand the paper better. On the other hand, the authors do not introduce the constraint solving problem. Moreover, they mention that solving such a problem is NP-hard as it involves the global minimization of a highly non-convex function but they do not explain or give an intuition on how the delta-complete algorithm can deal with this. The problem of finding complex non-linear Lyapunov functions together with the controllers for which they can guarantee stability is very important to the community and, therefore, this is a relevant paper.

Reviewer 3



This is a very strong contribution. In terms of organization, I would have moved all the numerical values learned (W_1, B_1, u_1, …) and many of the equations describing the systems the algorithm was evaluated on into the appendix, and spent more of the main paper describing more of the systems implications of the algorithm. In particular, Appendix B Table 1 and some commentary about how that cost differs from other approaches (LQR?) would be more immediately useful to the reader. I would also have loved to see validation of the approach on a real physical system with real-world complexity to help understand how this approach may scale. More discussion of the limitation of the approach would also strengthen the paper. None of these issues are critical. In general, making it possible to certify non-linear controllers is a huge problem, and new approaches to this issue are of general interest to the community. POST-REBUTTAL: thank you to the authors for their feedback. I had hoped for a slightly stronger statement than 'Yes, we *can* add evaluation of the control designs in physical wheeled robots.', but this doesn't change my overall assessment. Increasing my confidence level based on peer reviews.

[Author Response · NeurIPS 2019]

We thank all reviewers for the very important comments and suggestions for improving our paper.

The control problems in our setting may seem more basic than what is typically considered from a learning perspective. However, these are exactly the challenge problems that have occupied the control theory community for many decades and are far from being satisfactorily solved. For example: Tesla autopilot disclaims responsibility for turning safely in high speed; aircraft control needs to maintain very small angles-of-attack to avoid dangerous stalling; and in the recent DARPA challenge on robotics, all teams failed to robustly balance humanoid robots. These are direct results of failing to ensure large enough region of attraction when controlling nonlinear systems (at the scale of examples in our paper, with known dynamics). It is often claimed that these core nonlinear control problems can not benefit from recent progress in learning-based approaches, because of the need for precise global optimization. We counter this belief by showing that deep learning can deliver significantly better solutions than known methods, with provable guarantee. We are excited to see the power of neural networks in precisely capturing the nonconvex landscapes, and that constraint solving algorithms can rigorously verify global properties of these networks at the scale relevant to practical control.

Our approach is novel not just in the use of neural networks, but also in designing the combination of non-convex optimization methods to deliver precise global search required by the Lyapunov methods. Reviewer 2 questions the difference with [21]. Please note that [21] only estimates the ROA for a *given controller*, which is significantly simpler than our goal of *designing* controllers. Despite the easier goal, [21] only approximately models the ROA, treating it as a classification problem. There was no attempt in ensuring global properties of the networks. Moreover, [21] only works on the well-studied inverted pendulum example (on which we have reported a similarly-sized ROA, but with provable guarantee), and does not work on the other more complex examples in our paper. For *designing* controllers, existing methods (LQR and SOS) always rely on reduction to convex problems. For instance, SOS methods avoid full-scale global optimization by using polynomials that are positive-definite by design, at the cost of severely limiting the landscapes that can be captured. Our approach is the first that shows feasibility of using non-convex optimization methods and generic function approximators to rigorously satisfy the Lyapunov conditions. The control designs are strictly better than known solutions, for being more robust without using more complex control policy classes.

We agree with all reviewers that the numerical details of the experiments can be moved to the Appendix. We included them because the use of neural networks is very nonstandard in this setting, and we wish to give complete details of the learned results for easy validation of their correctness. We will expand explanations of the background and move algorithmic descriptions and more experiment statistics from the Appendix to the main sections.

**Reviewer 1: Will Lyapunov risk as cost function yield robust feedback controllers even in nonsmooth cases?** Very likely. In ongoing work we see benefits of minimizing Lyapunov risk in actor-critic RL, when the Lyapunov risk and policy gradients do not misguide each other. **Do ReLU networks work in practice?** Because of non-smoothness of ReLU, direct encoding does not work for the Lie derivatives. Approximation is possible, but we need to appropriately bound the encoding error. Note that the networks are small, and the choice of nonlinear unit does not affect training speed much. **Have you tried more complex systems with learned dynamics model or with uncertainty?** We can extend the examples with bounded noise and compare with LQ-Gaussian methods, although rigorous claims require extending the theory to systems with differential inclusions. **Please provide learning curves for the various experiments. How often does it fail?** Failure happens when verification takes too long. For instance, we can extend the humanoid model to more links; then the Lyapunov risk can still be minimized well, but the constraint solving time can increase exponentially, reflecting the inherent complexity of the problem. **Is there a connection to Contraction Theory?** Definitely. We see exciting possibilities for contraction theory based on neural network Lyapunov functions.

**Reviewer 2: Is the contribution simply in using a NN instead of other function approximators?** No. See Line 13 above. **Explain better the non-linear constraint problem and delta-complete algorithms.** We will add more background. You could consider the constraints as defining a multivariate nonlinear cost function, and the constraint solving problem is about finding its global minimum, which is NP-hard. **Clearer presentation of the experimental setting.** Yes, we will move more details to the main text. **State more clearly: known dynamics and small NN.** Yes, we will add to Line 19 in the introduction. **Reformulate bold claims.** Yes, we will remove speculations and focus on technical claims. **The final region of attraction does not cover the whole state space. This means that you cannot guarantee that the Lyapunov stability condition holds everywhere.** Incorrect. The Lyapunov conditions are guaranteed to hold everywhere within the entire red circle in the graphs (it is why the computation is hard). Falsification terminates for that entire region. Within this region, the ROA also needs to be fully contained in some level set of the Lyapunov function (Def 7). Thus, the ROA is always a proper subset of the fully-validated larger region.

**Reviewer 3: Commentary about how that cost differs from other approaches would be more immediately useful to the reader.** Yes, we will move Table 1 from the Appendix to the main text and add more columns. **I would also have loved to see validation on a real physical system with real-world complexity.** Yes, we can add evaluation of the control designs in physical wheeled robots. **More discussion of the limitation of the approach.** Yes, we will rewrite the introduction to improve clarity, by further elaborating on some points mentioned in this rebuttal.

[Meta-Review · NeurIPS 2019]

This paper proposes learning a Lyapunov function using neural networks for control of nonlinear dynamical systems. The reviewers found that the presentation of the paper, experimental tests and discussion of the generality of the method should be improved; however, they agreed that the paper was novel and would be of interest for the NeurIPS community.